# Rapid viscous flow of crustal rocks controls dyke emplacement in the ductile crust

Hans Jørgen Kjøll [1] ✉, Thomas Scheiber [2] & Olivier Galland [1]

Magmatic dykes are the main pathways for magma through Earth's crust. They are often assumed to be magma-filled fractures that propagate as thin, tapered sheet intrusions through mode I tensile opening at the fracture tip and elastic bending of the host rock along the dyke walls. Here, we present field evidence from northern Sweden, showing that deep-crustal dyke emplacement was, despite high strain rates, associated with significant ductile deformation of the host rocks. On average, 25% of the dyke thickness was accommodated by ductile flow of the immediate host rock. Modelling magma cooling times suggests average ductile strain rates of $10^{-3}$ s$^{-1}$ to $10^{-6}$ s$^{-1}$, i.e. 6 to 10 orders of magnitude faster than typical tectonic ductile strain rates in the middle crust. These results have major implications for understanding ductile crustal strength and the interpretation of geophysical signals used to mitigate geo-hazards in volcanically active areas.

Mafic sheet intrusions, such as dykes and sills, are the most fundamental pathways for magma transport through the Earth's crust and the main feeders of fissure eruptions[1-4]. Direct observations of dyke and sill propagation in the subsurface are challenging and limited to geophysical methods, constricting our understanding of the dynamics and physics governing their propagation and emplacement[5]. Dykes and sills exhibit low thickness-to-length aspect ratios from $10^{-2}$ to $10^{-4}$ [6-8], suggesting that they may be modelled as internally pressurized, planar cracks which propagate in a brittle, elastic solid by tensile mode I fracturing at their tips[1,6,9-11]. These models are based on the Linear Elastic Fracture Mechanics (LEFM) theory, which predicts that (1) dykes exhibit elliptical shapes, (2) their tips are sharp and tapered, and (3) the opening of the dyke is accommodated by elastic bending of the host rock. This theory is the foundation for interpreting geophysical[5] and geodetic[3] data associated with dyke emplacement in active volcanic regions.

Geological field observations, however, show that numerous dykes and sills do not have the elliptical shape predicted by the LEFM theory. Their cross-sectional shapes can range from parallel walls, except close to their tips (Fig. 1)[12], to complex thickness distributions including thickening near the tips[13,14], which is in disagreement with the predictions from the LEFM theory. In addition, the tips of numerous dykes and sills exhibit a wide range of shapes from sharply tapered, i.e.

in agreement with the LEFM theory (Fig. 1)[12,15,16], to blunt, rounded or even square, associated with elastoplastic and viscoelastic deformation of the host rock[16-21].

An argument used to justify the relevance of LEFM-based models applied to dykes and sills is that inelastic deformation occurs in a very small host rock volume at their tips, the so-called process zone, so that its effect may be negligible[10,11,22]. Nevertheless, Rubin[17] and Scheibert et al.[23] proposed an extended LEFM theory that accounts for plasticity at the intrusion tips and suggest that plastic deformation can have a significant effect on the intrusion propagation dynamics, even though it is restricted to process zones of negligible size. However, these models assume that inelastic deformation occurs only at intrusion tips and assume that only elastic deformation occurs along the intrusion walls.

The LEFM assumptions mostly apply to dykes and sills emplaced in the upper crust where rocks are expected to deform in a brittle and elastic manner (Fig. 1). Dykes and sills are, however, also common in lower crustal levels, where ductile deformation regimes are dominant[20,24]. The extent to which ductile deformation accommodates, at least partly, dyke emplacement both in the brittle-ductile transition zone and in the ductile crust is currently poorly known. Tectonic ductile strain rates are typically on the order of $10^{-12} - 10^{-15}$ s$^{-1}$ [25,26], whereas strain rates associated with magma emplacement are expected to be much higher at around $10^{-2} - 10^{-13}$ s$^{-1}$ [27]. Assuming a Maxwell

[1]Njord centre, Department of Geosciences, University of Oslo, Oslo, Norway. [2]Department of Civil Engineering and Environmental Sciences, Western Norway University of Applied Sciences, Sogndal, Norway. ✉e-mail: h.j.kjoll@geo.uio.no

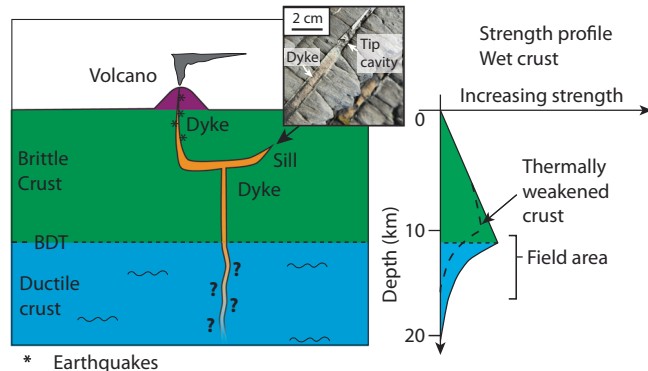

**Fig. 1 | Conceptual model of dyke emplacement across the brittle–ductile transition.** Schematic representation of dyke emplacement across the brittle-ductile transition (BDT). Dyke propagation and emplacement mechanisms are well described in the brittle crust, but remain poorly constrained in the ductile crust.

rheology − a material model corresponding to a viscous damper and an elastic spring mounted in series − for lower crustal rocks[17,28], these high strain rates could justify the brittle emplacement mechanisms assumed in the LEFM model (Fig. 1). This assumption, however, has so far not been validated in the field.

In this contribution, we present field observations from an exhumed dyke complex of a magma-rich rifted margin, exposed in the Scandes of northern Sweden. The dyke complex was emplaced in thermally weakened marbles and arkoses. Folds in the weak host rocks, with axial planes sub-parallel to the dyke contacts, are documented along the dyke walls. Our observations show that 25% on average of the dyke thickness was accommodated by ductile deformation and flow of the host rock along the dyke walls. We constrain the strain rate for this ductile deformation to be consistently $10^{-3}$ to $10^{-6}\,s^{-1}$, i.e., 6 to 10 orders of magnitude faster than average tectonic ductile strain rates. This shows that significant inelastic deformation is accommodated during dyke emplacement, including deformation along dyke walls, which is contrary to existing model predictions. Further, we show that this ductile deformation may be extremely fast in the ductile crust.

## Results
### Geological setting
The study area is located in Sarek National Park, Northern Sweden, and hosts a succession of Neoproterozoic meta-sedimentary rocks invaded by abundant Ediacaran-aged mafic dykes (Fig. 2). The studied rocks formed part of the outermost margin of the paleocontinent Baltica (northwestern Eurasia) and underwent subduction early in the Caledonian Orogeny[29–31]. These rocks were later exhumed and thrust onto Baltica as part of the regional Seve Nappe Complex during the Silurian-aged Scandian phase (continent-continent collision) of the orogeny[30,32–38]. Despite the complex tectonic history of the Seve Nappe Complex, it hosts numerous (km-scale) megalenses, that escaped penetrative Caledonian strain. In these lenses, the original magmatic and sedimentary relations are preserved, making them a window into the deep parts of a rifted margin and enabling the study of dyke emplacement mechanisms in the ductile crust.

The dyke complex in the Sarek area is hosted by three successive sedimentary formations (Fig. 2D)[39–41]. In this contribution, we study an outcrop at the Favorithällen locality (Fig. 2B), where dykes were emplaced within the Neoproterozoic Spika Formation[39,42]. It is characterized by thinly bedded (5–30 cm) carbonates interbedded with thin (< 1 cm to 5 cm) calc-silicate beds. Sedimentary structures, such as dewatering structures and cross-bedding are preserved within these rocks[41]. The presence of scapolite and magnesite has been interpreted to represent former evaporite deposits[41] and recently described

stromatolites underpin the interpretation of a shallow deposition level of the Spika Formation[39]. The sedimentary basin was formed as a precursor to the Ediacaran continental breakup[43,44]. Prior to breakup the succession was affected by layer-parallel stretching, which is recorded as boudinage of the relatively strong siliciclastic layers[40,41].

During Ediacaran continental breakup, marking the inception of the Iapetus Ocean[20,35,40,44,45], the sedimentary succession was intruded by a network of anastomosing, geochemically enriched mafic dykes, at ca. 608 Ma[44] (Fig. 2E). This dyke complex has been interpreted as an analog to the outermost margin or the ocean-continent transition of a magma-rich rifted margin[29,40,46]. In Sarek, the dykes constitute between 70 and 100% of the area and locally form sheeted-dyke complexes[35,40,42,44]. Dyke thicknesses have been documented to range from <1 m to 17 m and follows a Weibull distribution with a mode of ca 3 m[20].

A high geothermal gradient was established for the area as the dyke complex was emplaced at c. 4.5 kbar, and the host rock temperature was high and locally near the wet granite solidus at around 700 °C ± 25 °C[40]. Dyke geometries and structures suggest that the ambient conditions went from brittle to ductile deformation during emplacement of the dyke complex[20]. In addition, it has been shown that the rate of magma input was greater than the rate of tectonic extension, leading to crustal thickening in Sarek during emplacement of the dyke complex[20].

### Results from the Favorithällen locality
The Favorithällen locality (Fig. 2B) exposes a glacially polished section of doleritic dykes hosted by the Spika Formation, which allows the study of dyke geometries and the deformational structures in the host rock. The following sections present and document how host deformation accommodated the emplacement of mafic dykes in this deep crustal section.

The bedding of the metasedimentary strata generally strikes WNW-ESE and is subvertical to steeply-dipping (Fig. 3). The dykes can be grouped into two orientation sets, (1) E-W striking and dipping between 29° and 69° towards the S, and (2) NNE-SSW striking and dipping between 27° to 68° towards the ESE (Fig. 3B). The dykes in the study area range in thickness from 5 m to <1 m with an average thickness of c. 1.5 m, measured orthogonal to the dyke contacts. The dykes commonly show emplacement-related features such as stepped contacts with small offshoots (horns), chilled margins and broken bridges (Figs. 3, 4). Stepping of the dyke-host contacts are often located at bed boundaries or where the proportions of calc-silicate beds change with respect to the marble beds (Fig. 4C).

### Deformation features near dyke walls
Folding of the host strata is observed along numerous dyke-host contacts (Fig. 5). The folding resembles buckling and is locally disharmonic and chaotic where marble beds are intercalated by thin calc-silicate beds (Fig. 5B, D). The folding intensity decreases away from the contacts (i.e. amplitude decreases, and wavelength increases) over a relatively short distance of <1 m (Fig. 5). The wavelength of the folding ranges from centimeters to decimeters (Fig. 5). Locally, neighboring folds have opposite facing directions (Fig. 5B and D), suggesting that shortening is accommodated by pure shear. The interlimb angles vary significantly and range from >70° to <30° depending on the distance from the dyke contact and the thickness of the folded calc-silicate layer (e.g. Fig. 5C). The axial planes of the folds are moderately dipping to the SSE and to the ESE (Fig. 5E). These two axial plane orientations are similar to the two orientations recorded for the contacts between the dykes and the host rock (Fig. 3B). Locally, folding is asymmetric such that the host rock is folded on one side of the dyke and not the other (Fig. 5B).

Figure 5E shows a remarkable correlation between stepping of the dyke and the style and intensity of folding of the adjacent host rock strata. Where the dyke steps into the host rock, the folding intensity is

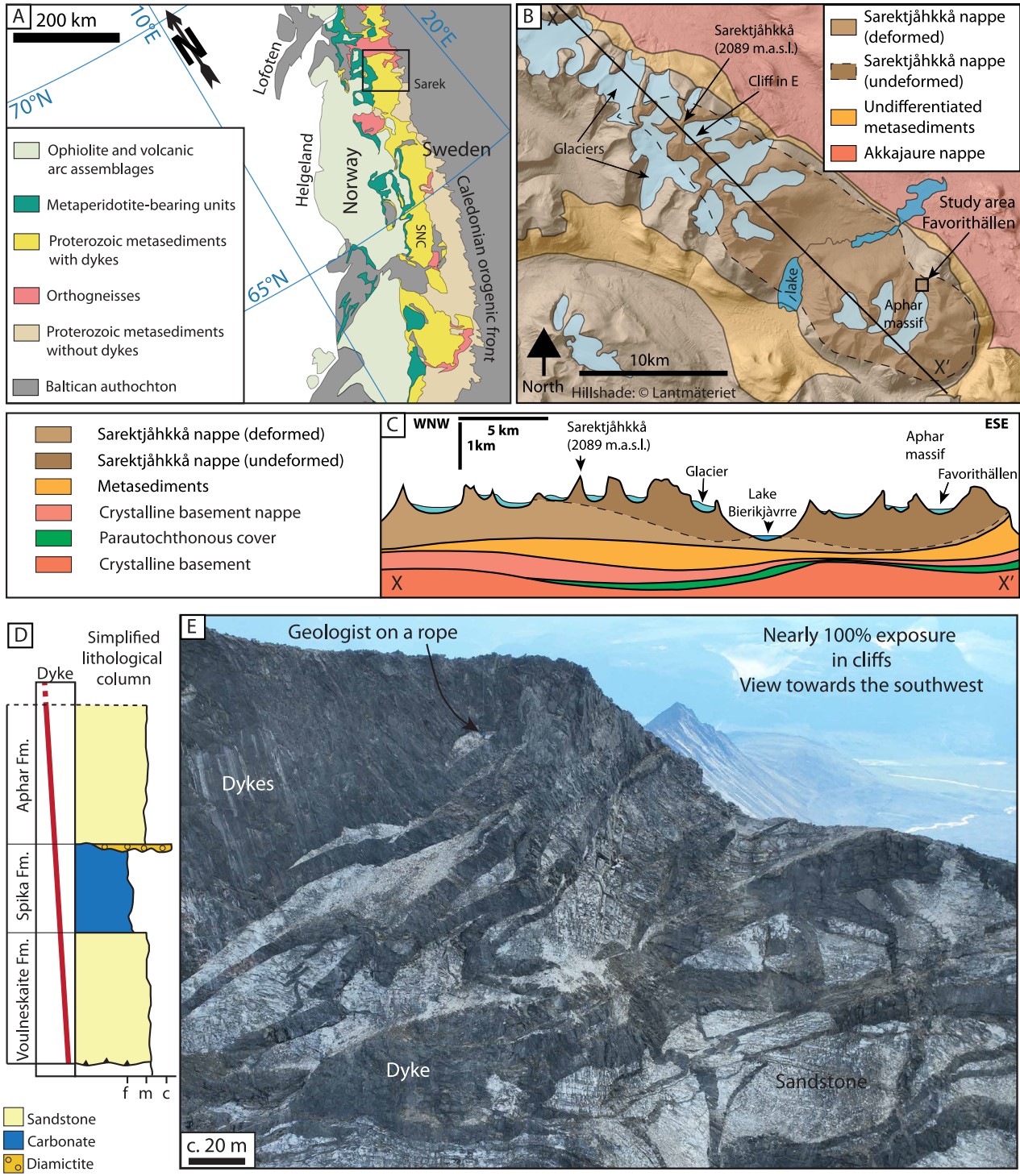

**Fig. 2 | Geological setting of the study area. A** Simplified tectonic map of the Central Scandinavian Caledonides in Northern Sweden and central Norway modified from[38]. Black rectangle shows the approximate location of the National Park. **B** Geological map after[42] of the study area and the Sarektjåhkkå Ridge, a well-preserved lens within the Seve Nappe Complex that escaped Caledonian deformation. Base map hillshade is from the Swedish map authorities, © Lantmäteriet.

**C** Geological cross section after[64] along the Sarektjåhkkå and Aphar Ridge (X-X' cross section in **B**). **D** Simplified stratigraphic column, modified from[39], showing the three different sedimentary formations in Sarek. **E** Oblique drone photograph of the dyke complex in a glacial cirque, displaying the intricate plumbing system preserved in Sarek.

higher, indicating that ductile folding accommodated enhanced dyke inflation. The folding style and intensity are largely controlled by the host rock lithology (Fig. 5E), where the non-folded host rock consists of a calc-silicate-dominated lithology, and the folded domain is marble-dominated. This relationship shows how the host rock lithology affects the deformation style, which, in turn, affects the dyke thickness and morphology. Further, the folded layers display progressive folding of the host rock instigated by short-wavelength (<1 cm) buckling of the thin calc-silicate layers that progresses into buckling with a much longer wavelength (>10 cm) (Fig. 5E).

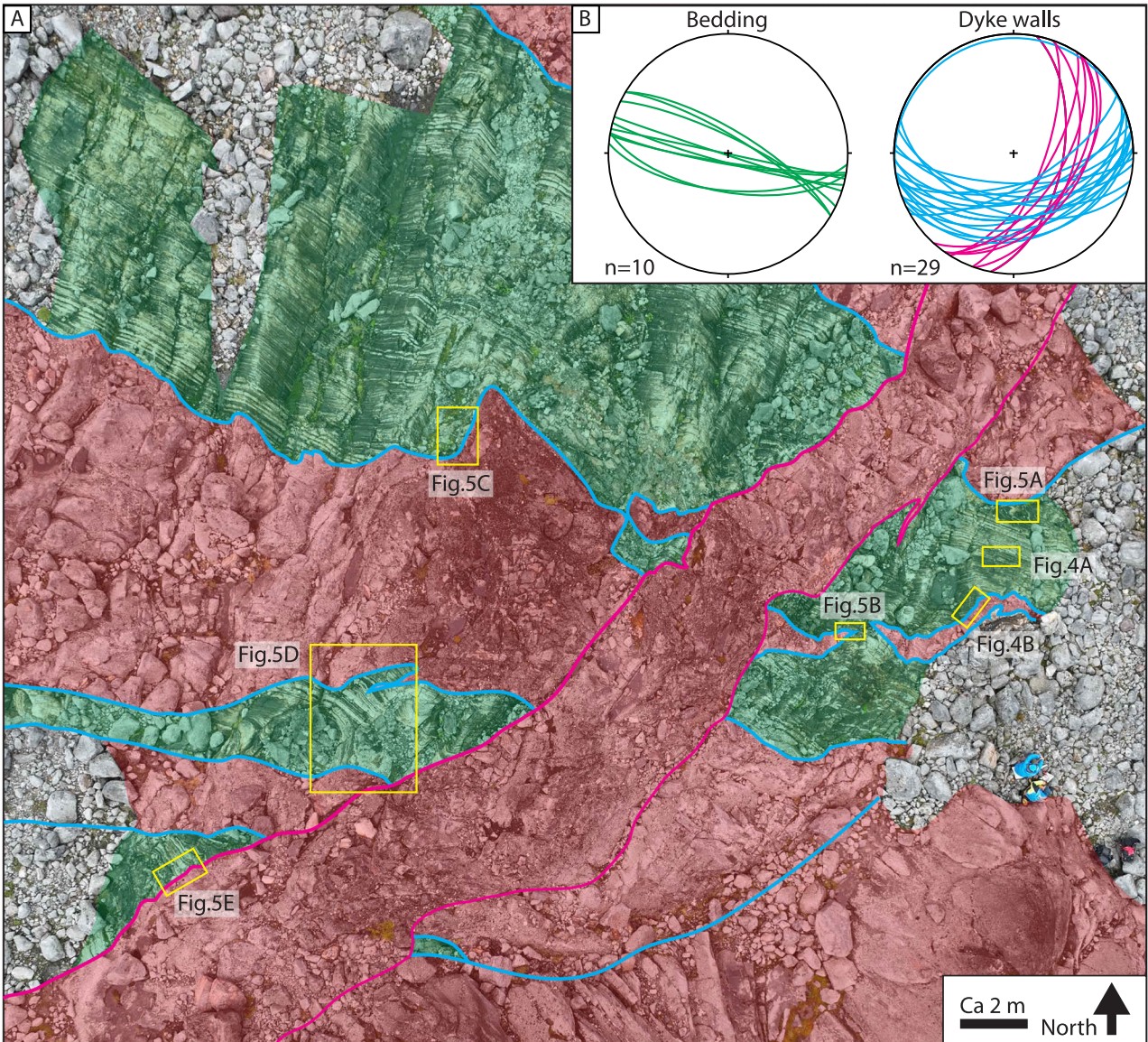

**Fig. 3 | Structural relationships between dykes and host rocks at the Favorithällen locality. A** Drone photograph from the central part of the Favorithällen locality showing host rock (green) and dyke (red) relationships. A NNE-SSW striking dyke crosscut E-W striking dykes in this locality. Yellow rectangles highlight locations of Figs. 4, 5. **B** Equal-area lower hemisphere stereoplots of sedimentary bedding (green) in the host rock and dyke walls from Favorithällen. Note consistent bedding orientation and two sets of dykes, E-W striking (blue) and NNE-SSW striking (magenta).

Locally, folding and ductile flow of the host rock is accompanied by nearby thin, ductile shear zones, which crosscut sedimentary beds and accommodate displacements on the cm-scale (blue line in Fig. 5B). This suggests that shortening occurred perpendicular to the dyke contacts.

In order to assess the distribution and style of deformation in the host rock, we investigated the calcite microstructures in the deformed contact zone and compared them to the undeformed host rock away from the dyke contact. The microstructures at the contact include undulatory extinction, twinning and subgrain formation (Fig. 6A). Locally the subgrain boundaries are offset by twin boundaries (Fig. 6A). In many instances, the twins crossing the subgrain boundaries are also bent and/or kinked, indicating the rotation of the crystal lattice by dislocation movement[47] (Fig. 6A). The twin density close to the contact is high (Fig. 6A). Away from the deformed contact zones, however, the calcite crystals are relatively strain free with little evidence for twinning, i.e. the twin density is low compared to the sample from close to the contact (Fig. 6B). In addition, calcite crystals are present as inclusions within the contact metamorphic garnets (Fig. 6B).

The preserved cross-bedding in the sedimentary host rocks, the magmatic textures in the dykes, together with the preserved primary emplacement structures show that the studied area has experienced negligible Caledonian deformation, if any. These structures must therefore record deformation associated with dyke emplacement, rather than later Caledonian events. However, considering (1) the systematic folding concentrated at a short distance (<1 m) from the dyke walls, (2) the increased intensity of the deformation recorded in the microstructures towards the dyke contacts, (3) the correlation between orientations of the fold axial planes and the dyke walls, (4) the local correlation between the folding style/intensity and the local intrusion morphology, and (5) the correlation between intrusion morphology and local ductile shear zone geometries strongly suggest that the observed ductile folding results from, and partly accommodates, the emplacement and growth of the dykes.

The exact timing of the deformation is difficult to constrain, but the observations of folding along the dyke walls close to dyke tips (Fig. 5B) as well as the correlation of dyke step geometries and folded host rock

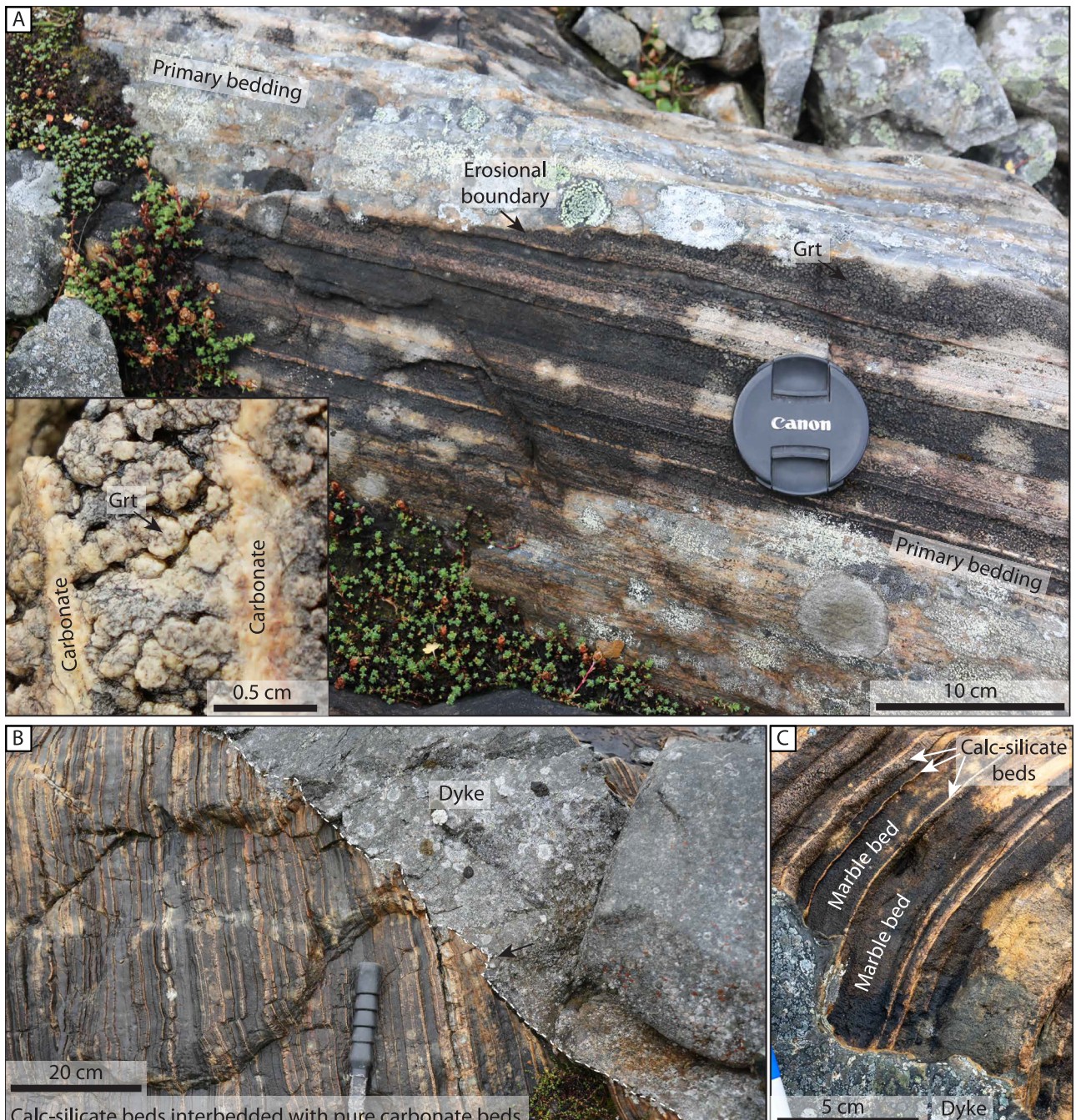

**Fig. 4 | Preserved primary structures in the metasedimentary host rock of the study area. A** Primary sedimentary structures, i.e. low angle erosional boundary in cross-bedded strata. Note that strata directly below the erosional boundary is completely statically transformed to garnet as expressed by the nodular texture. Inset shows a close-up of the nodular texture consisting of Ca-rich garnet porphyroblasts. **B** Dyke cutting the calc-silicate marble without accommodating internal deformation. Note the highly irregular contact which locally follows bed boundaries. **C** Close-up of dyke contact which steps when it intersects a calc-silicate bed.

(Fig. 5E), suggest that the folding occurred early in the dyke inflation event while a significant part of the dyke interior was still molten.

### Ductile strain rate estimate

Since the ductile deformation of the host rock near the dyke contacts is related to the dyke emplacement (Fig. 5), we consider ductile strain rates at the time scale of dyke emplacement. To quantify the ductile strain rate, we estimate independently (1) the ductile strain accommodated by folding and (2) the time scale of dyke emplacement and cooling.

We estimated the ductile strain using equation:

$$\varepsilon = \left(\frac{l_0 - l_1}{l_0}\right) \tag{1}$$

where $\varepsilon$ is the longitudinal strain, $l_1$ is the distance from the dolerite-host rock contact to where the deformation fades and $l_0$ is the length of the folded beds measured from the dyke contact to the same point as for $l_1$ (see Fig. 9 in methods). Measurements conducted on scaled field photographs yield shortening strain in the range of

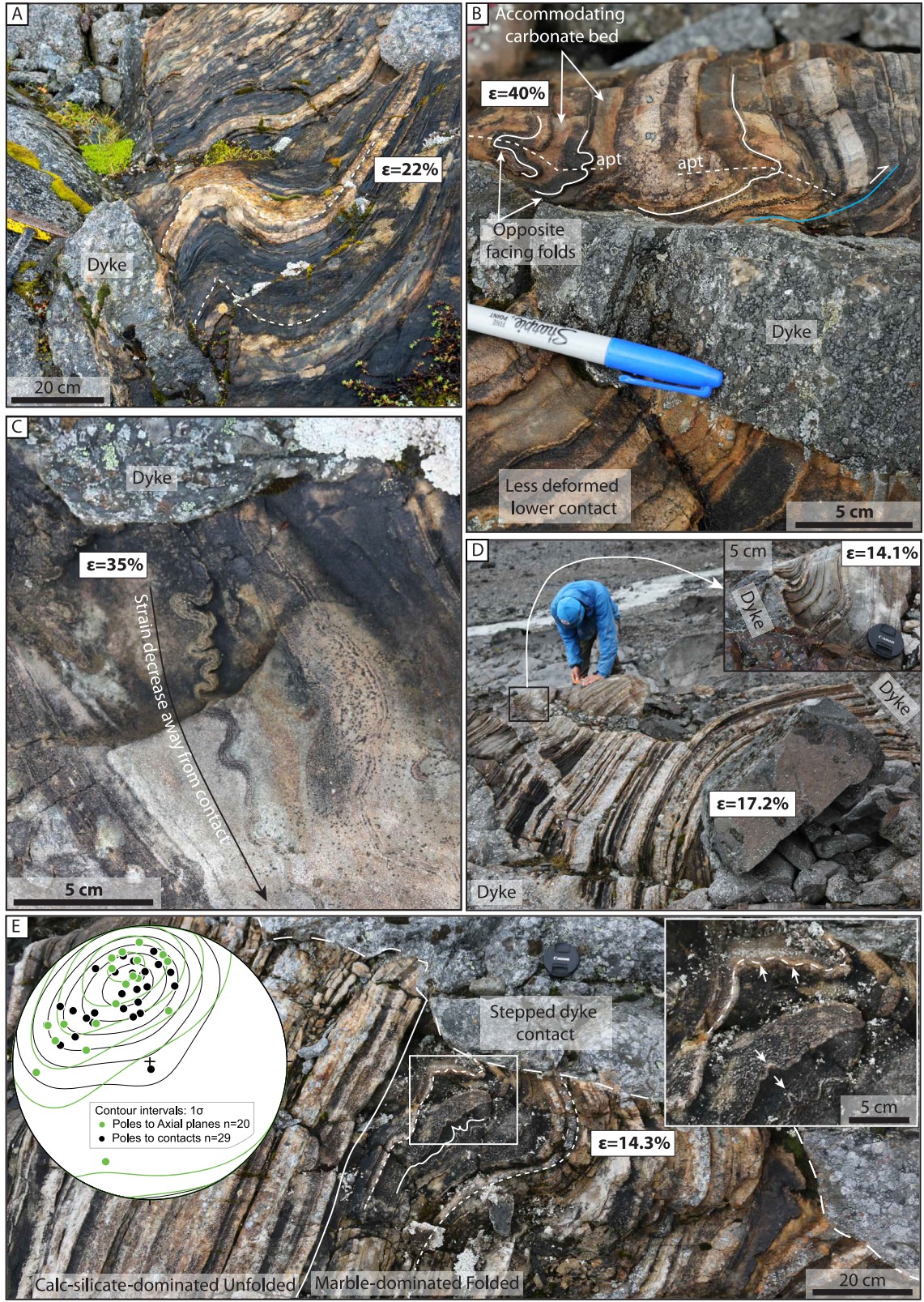

14-40 % with an average of 23 % ($\sigma = 8.6$; $n = 11$; Fig. 7; Table 1). These measurements are from photographs taken parallel to the fold axes.

When correlating the amount of shortening with the actual dyke thickness adjacent to the folds, it becomes evident that the total shortening accommodated by folding generally represents between 2.7-65.2 % of the total dyke thickness with an average of 25% (Fig. 7 and Table 1). Thin dykes, such as the dyke in Fig. 5B, has the highest amount of strain compared to the total thickness (Table 1).

To estimate the time scale of the observed ductile deformation, we assume that the folding occurs at similar time scales as dyke cooling, given the close link between the geometry of the dyke contact and the folding of the host rock (Fig. 5E). Most of the dykes in the study area are 0.1-8 m thick but the thickest dyke with

**Fig. 5 | Folding and ductile deformation of host rocks adjacent to dyke contacts. A** Folding of heterogeneous metasedimentary strata shows that the strata are folded close to the dyke, then the folding is rapidly dissipating away from the dyke contact. **B** Significant folding of marble and thin calc-silicate beds along the upper contact to the dyke. Note that the folds in the calc-silicate beds have opposite-facing directions, suggesting that disharmonic folding formed during pure shear deformation of the host rock. The asymmetry of the system is likely related to the highly heterogeneous nature of the system. Apt: axial plane trace. **C** Buckled, thin bed of calc-silicate in a thicker unit of marble showing a decrease in strain with increasing distance from the contact. **D** Meter-scale open fold between two dykes and cm-scale folding next to a dyke offshoot (inset) **E** Dyke steps from calc-silicate dominated (strong) to marble dominated (weak) lithologies. The marble dominated lithology is folded. The insert shows details of the folded layer, displaying buckling, which is re-folded by a chevron-type fold. Lower hemisphere equal-area stereoplot showing poles to measured fold axial planes (green) and dyke walls (black), and pole density contour lines. Note the similar orientations suggesting a genetic and temporal link between folding and dyke emplacement. See locations of sub-figures in Fig. 3. $\varepsilon$ is the longitudinal strain defined in Eq. 1 and used in the calculation of the strain rate below.

measured folds is 5 m thick (Fig. 3; Table 1). Characteristic cooling times ($t$) for dykes with varying thicknesses (0.05-11 m) at various host rock temperatures (600-800 °C) are shown in Fig. 7.

Consequently, estimates of the strain rate ($\dot{\epsilon}$) can be determined using the formula:

$$\dot{\epsilon} = \left(\frac{\varepsilon}{t}\right) \tag{2}$$

where $\varepsilon$ is the longitudinal strain, as given above, and t is the time it takes for a dyke of various documented thicknesses to solidify at various host rock temperatures (Table 1). For an ambient temperature of 600 °C, it would have taken between 0.05 days (4240 seconds) and 145 days ($1.25 \times 10^7$ seconds) to crystallize 0.1 to 5 m thick dykes, respectively (Fig. 7C). This yields strain rates ranging from $9.42 \times 10^{-3}$ s$^{-1}$ to $2.67 \times 10^{-6}$ s$^{-1}$ (Table 1). When considering a host rock temperature of 800 °C the cooling time increases and range from 0.11 days ($9.23 \times 10^3$ seconds) to 315 days ($2.72 \times 10^7$ seconds; Fig. 7C). The strain rates range from $3.66 \times 10^{-3}$ s$^{-1}$ to $1.23 \times 10^{-6}$ s$^{-1}$ (Table 1).

## Discussion

The excellent and vast exposure of the dyke complex in Sarek provides an important window into dyke emplacement processes in the ductile crust. Most dyke propagation models consider dykes as fluid-filled mode I (tensile) fractures that form penny- or blade-shaped cracks with a thickness governed by the host rocks elastic properties[1,6,11]. The models also focus on the tip-processes, as this is where the stresses are highest[1,6]. Our observations are at odds with these models as they show inelastic deformation occurring along the dyke walls rather than at the dike tip. These models, therefore, require refinement in order to account for the effect of inelastic processes and deformation related to inflation of the dykes at deep crustal levels.

Elastic models are often utilized for host rock deformation during dyke emplacement[11,48], but the observations of small-scale inelastic deformation directly related to dykes and sills emplaced in sedimentary basins have shown that inelastic properties can be significant[21,49]. Undeformed and well exposed outcrops of dykes emplaced in deeper sections of the crust, where ductile deformation mechanisms prevail, are less common, and inelastic deformation related to the emplacement in the form of ductile flow of the host rock has so far not been documented. The data presented herein documents that between 2.7% and 65.2% of the dyke thickness was accommodated by folding of the host rock (average 25%; Fig. 7). The remaining was likely accommodated by other mechanisms such as localized ductile shear zones and distributed ductile flow within carbonate layers, as well as elastic deformation of the host or tectonic stretching. These findings show that, for weak lithologies, inelastic deformation of the host rock plays a previously underestimated role in dyke emplacement in the lithosphere, especially under conditions where the temperature is close to or above the brittle-ductile transition. Furthermore, these strains are significantly higher (Table 1) than what is assumed in the LEFM models, which assume small strains <1 %[50].

Dyke emplacement resulted in a compressional stress field that caused local folding of the host rock at the dyke contacts, behind the tip. One question that arises is the timing of folding with respect to the dyke emplacement: did the folding occur during fracture propagation, i.e., at the tip of the dyke, or along the dyke wall as the dyke was inflating? Our field observations show that the folds often are symmetrical and disharmonic with axial planes oriented sub-parallel to the dyke contacts, regardless of the orientation of the dyke (Fig. 5E), i.e., consistent with pure shear conditions. The observed shear zones are interpreted as resulting from superimposed simple shear in an overall pure shear stress field and are compatible with dyke-perpendicular shortening. The occurrence of one-sided folding of the host rock is likely controlled by local stress disturbances caused by the inherently heterogenous nature of dyke emplacement processes[51]. We therefore propose that the dyke tip propagated as a mode I fracture and that the folding occurred along the dyke wall during dyke inflation (Fig. 8). Inelastic processes are thus important not just at the tip of the propagating dyke but also along the dyke walls.

Similarly, we observe that during the early phases of dyke inflation, i.e. where the dykes are thin, folding accommodates nearly the entire dyke thickness, whereas for thicker dykes the observable accommodated strain is much less compared to the dyke thickness (Fig. 7). This could suggest that a significant amount of the host rock deformation may have occurred already during the early phases of inflation.

The observed strain gradient decreasing away from the dyke walls may be caused by dissipating stresses away from the dyke contact and thus the stresses exerted by the inflating dyke are accommodated by folding and ductile flow, or alternatively, the strain gradient reflects the thermal structure imprinted by the doleritic dyke on the host rock. In the latter alternative, the rheology would be weakest closest to the dyke where the temperature was highest, and would be stronger away from the dyke until it reaches the background temperature of the area. The high twin density and glide-controlled calcite microstructures, such as twinning and lattice distortion, observed at the dyke contact (Fig. 6A) indicate high strain rates with limited recovery[47,52,53]. Both the mechanical properties and thermal imprint on the host rock likely played a role in the deformation, but the contribution of these factors is beyond the scope of this study; rather, it is important to highlight that these processes acted in concert, affecting the deformation of the host rock.

We thus envisage an emplacement model where the dyke propagates at high strain rates by mode-I fracture through the hot ductile crust (Fig. 9A). This brittle fracture mechanism is accommodated by the rapid propagation of the fracture tip, embrittling the overall ductile host rock. The magma pressure pushes on the dyke wall, which flows in a ductile manner where the host rocks are weak (here marble; Fig. 8B). Where the host rock is strong, i.e., dominated by calc-silicate beds, the host rock resists magma intrusion (Fig. 8B and C). Continued inflation exaggerates the steps created by the change in host lithology strength and refolds some of the early folds (Fig. 8C). The thermal softening of the host may lead to disharmonic folding of the stronger layers accommodated by viscous flow in the weaker layers with axial planes sub-parallel to the dyke contact (Fig. 8C).

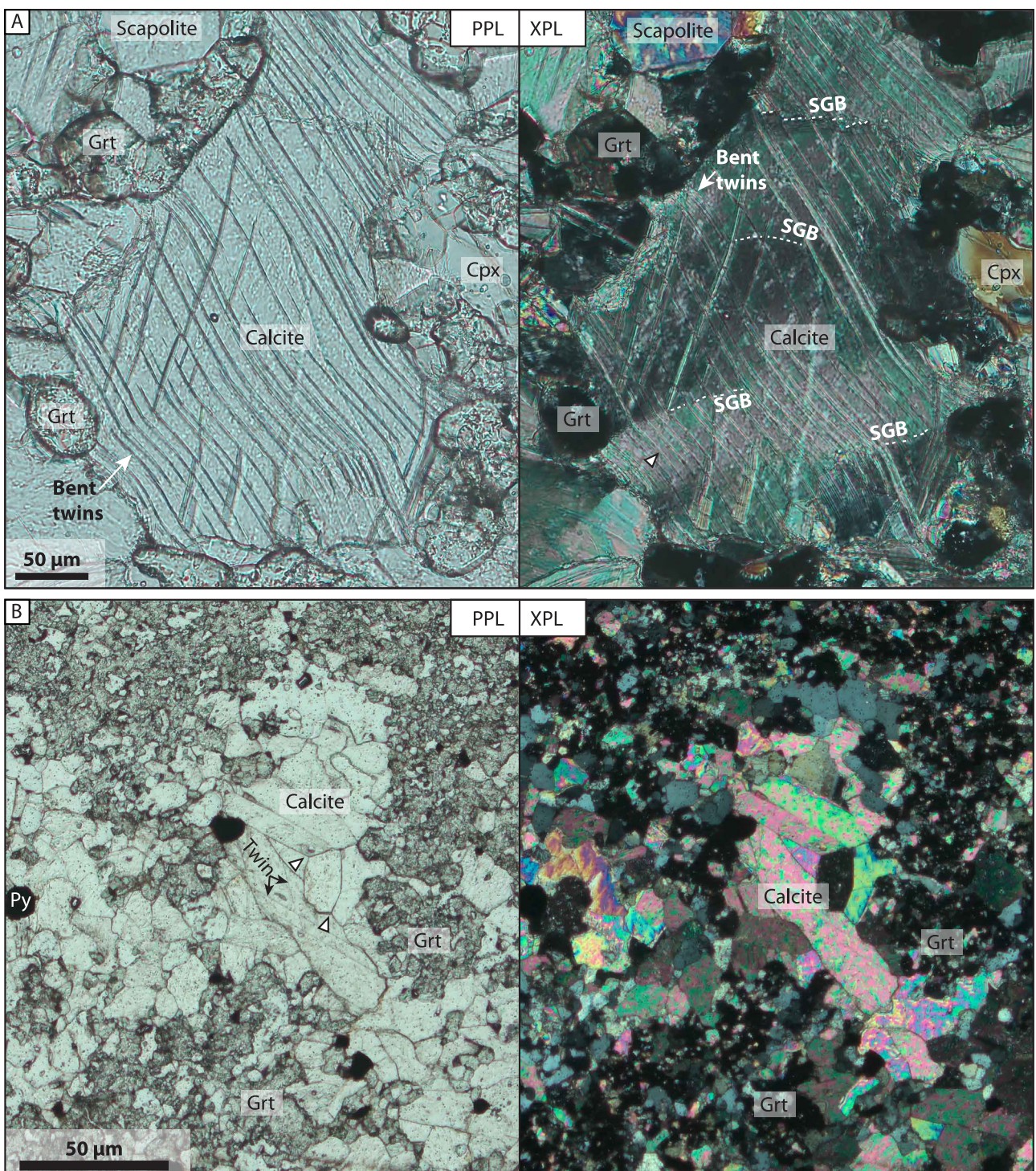

**Fig. 6 | Microstructural evidence of ductile deformation in calcite near dyke contacts.** Plane-polarized light (PPL) on the left and cross-polarized light (XPL) on the right **A** Sample from folded domain less than 5 cm from the contact. Carbonate crystal shows intracrystalline deformation through the presence of twins, lattice distortion and subgrain boundaries. Subgrain boundaries are highlighted by white dashed lines. Note that the subgrain boundaries are locally affecting the twin boundaries, but also the other way around where twin boundaries offset subgrain boundaries. **B** Sample from unfolded domain, ca 2 m away from dyke. A calc-silicate bed displaying garnet and elongated subhedral carbonate crystals. Some pyroxenes are also present in the garnet-rich domain. The texture shows static growth of the minerals with no evidence of intracrystalline deformation. SGB subgrain boundaries, Grt garnet, Cpx clinopyroxene, Py pyrite.

The strain rate calculations depend on the time it takes for the magma to crystallize, which we have modeled to be less than 315 days for the dykes and the thermal conditions in Sarek (Fig. 7C). This is a maximum estimate if we infer that the recorded folding of the host rock occurred during the inflation of the dyke. However, considering post-emplacement stress recrystallization processes it would represent a minimum estimate. A significant impact from post-crystallization stress relaxation is, however, less likely as we have documented a clear relationship between the geometry of the dyke and the folding as well as a clear asymmetry of the deformation where one side of the dyke is deformed, and the other is not (Fig. 5B). This shows that the deformation must have occurred while the dyke

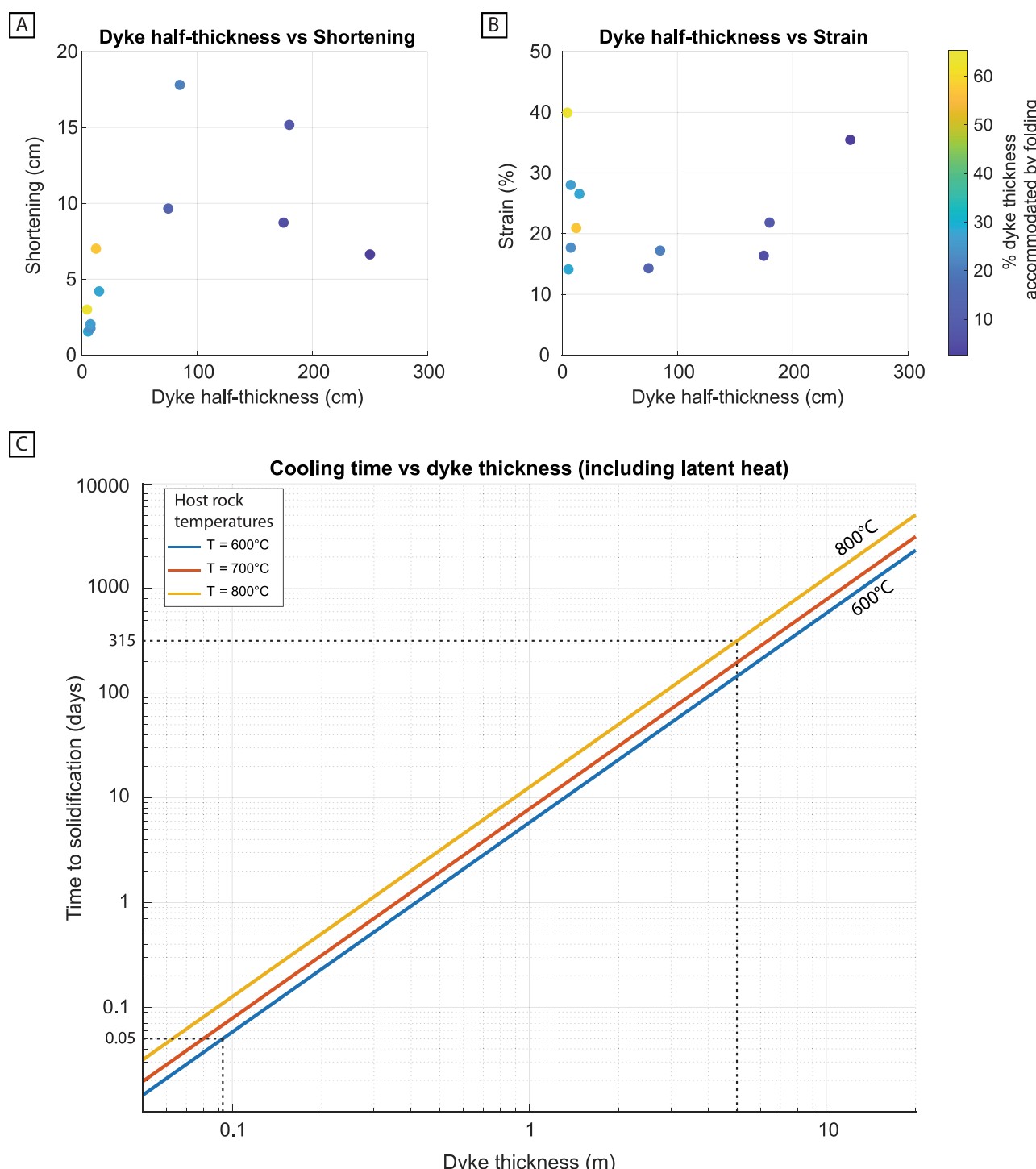

**Fig. 7 | Deformation measurements and thermal-cooling models for deep crustal dykes. A** shortening ($L_0$-$L_1$) vs the half-thickness of dykes and **B** longitudinal strain $\varepsilon$ vs the half thickness of dykes. The datapoints are color-coded according to the percentage of dyke thickness accommodated by folding. **C** Thermal modeling results showing the time it takes to reach the solidus temperature for mafic dykes of different thicknesses and at varying host rock temperatures. The modeling assumes a magma starting temperature of 1250 °C[40,45]. Solidus temperature magma: 1000 °C[65], Ambient temperature: 600-800 °C[40], magma liquidus temperature: 1300 °C[66], Dyke thicknesses 0.45-5 m (Table 1), Thermal conductivity: 2.6 W m$^{-1}$ K$^{-1}$ [67], Density: 2800 kg m$^{-3}$ [67], Specific heat capacity 1480 J kg$^{-1}$ K$^{-1}$ [67], Latent heat of fusion 400 kJ kg$^{-1}$ [67].

remained at least partially liquid excluding a significant contribution of post-crystallization stress relaxation. We therefore conclude that the modeling provides a useful maximum estimate for the timeframe of when the deformation occurred.

Average tectonic strain rates during extension of the ductile middle crust typically range around $10^{-12}$ – $10^{-15}$ s$^{-1}$ [25,26] but are sensitive

to the strength of the crust which, in turn, depends on the input of magma[13,54]. Our observations and calculations from dyke intrusion-related ductile deformation indicate strain rates in the order of $10^{-3}$ – $10^{-6}$ s$^{-1}$ and are as such remarkably high compared to typical tectonic strain rates in the ductile crust. The main cause of the fast strain rates is the geologically speaking quick emplacement and cooling of dykes,

**Table 1 | Results from the fieldwork together with calculations of strain, cooling time and strain rate**

| Photo# | Shortening (cm) | ε (%) | Dyke half thickness (cm) | % of dyke thickness | Time (s) 600°C | Time (s) 800°C | Strain rate (s⁻¹) 600°C | Strain rate (s⁻¹) 800°C |
|---|---|---|---|---|---|---|---|---|
| Dyke1 Fig. 5E | 6.6 | 35.5 | 250 | 2.7 | 1.25E + 07 | 2.72E + 07 | 2.83E-06 | 1.30E-06 |
| Dyke2 Fig. 5C | 9.7 | 14.3 | 75 | 12.9 | 1.13E + 06 | 2.45E + 06 | 1.27E-05 | 5.83E-06 |
| Dyke3 Fig. 5B | 3.0 | 39.9 | 4.6 | 65.2 | 4.24E + 03 | 9.23E + 03 | 9.42E-03 | 4.33E-03 |
| Dyke4 | 4.2 | 26.6 | 15 | 28.0 | 4.51E + 04 | 9.81E + 04 | 5.89E-04 | 2.71E-04 |
| Dyke5 | 8.7 | 16.4 | 175 | 5.0 | 6.14E + 06 | 1.34E + 07 | 2.67E-06 | 1.23E-06 |
| Dyke6 Fig. 5D | 17.8 | 17.2 | 85 | 21.0 | 1.45E + 06 | 3.15E + 06 | 1.19E-05 | 5.46E-06 |
| Dyke7 Fig. 5D | 1.6 | 14.1 | 5.5 | 28.2 | 6.06E + 03 | 1.32E + 04 | 2.33E-03 | 1.07E-03 |
| Dyke8 | 1.7 | 17.7 | 7.5 | 23.5 | 1.13E + 04 | 2.45E + 04 | 1.57E-03 | 7.21E-04 |
| Dyke9 | 2.0 | 28.0 | 7.5 | 27.1 | 1.13E + 04 | 2.45E + 04 | 2.49E-03 | 1.14E-03 |
| Dyke10 | 7.0 | 20.9 | 12.3 | 57.1 | 3.03E + 04 | 6.60E + 04 | 6.90E-04 | 3.17E-04 |
| Dyke11 Fig. 5A | 15.2 | 21.8 | 180 | 8.4 | 1.62E + 06 | 3.53E + 06 | 1.34E-05 | 6.18E-06 |
| Average: | | 23 | 74 | 25 | | | 1.56E-03 | 7.16E-04 |
| Max | | 39.94 | | 65.2 | | | 9.42E-03 | 4.33E-03 |
| Min | | 14.11 | | 2.7 | | | 2.67E-06 | 1.23E-06 |

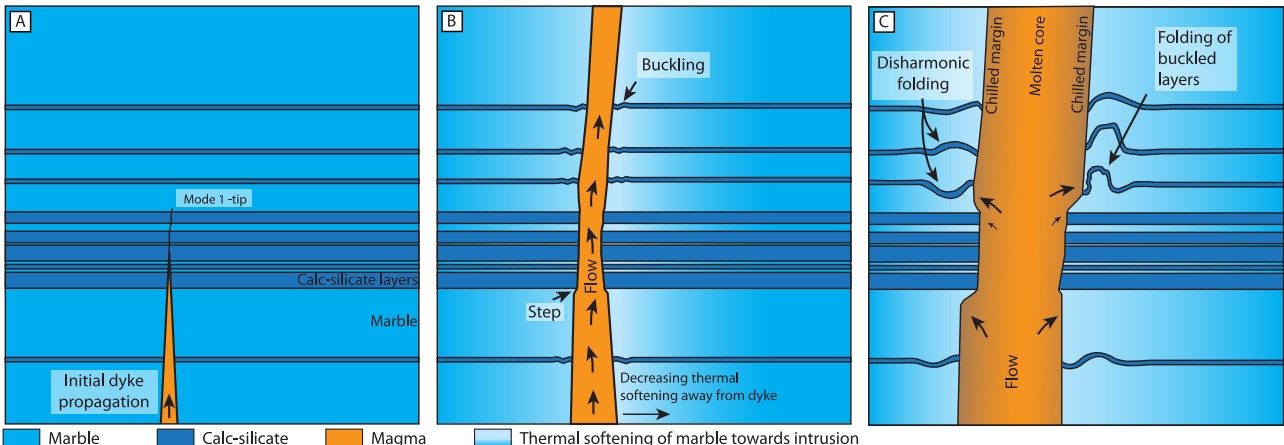

**Fig. 8 | Conceptual model of dyke emplacement and host-rock deformation in the ductile crust. A** Initial fracture propagation driven by magma pressure as mode I fracture. **B** Magma ingress and start of inflation leads to buckling of thin strong calc-silicate layers. Buckling is accommodated by viscous flow in the weak marble host as shown by the gradient towards lighter blue color. Steps in the intrusion wall develop where host rock is dominated by strong lithologies. **C** Continued inflation of the dyke leads to folding of the buckled layers. Since the carbonate-dominated lithology is weaker than the calc-silicate-dominated lithology, it can accommodate more shortening and thus the steps grow. Disharmonic folding of thin calc-silicate beds indicates the overall weak nature of the host rock.

which in the study area range from 315 days to less than a day. Similar and even faster strain rates ($10^{-2}$ to $10^{-12}$) have been calculated for granitic systems, but these systems are more complex and the deformation is caused by the emplacement of a granitic pluton[27]. The presented data is of a simpler system of single dykes deforming the host rock to accommodate their growth. In addition, the Sarek host rock consists of layers with a significant competence contrast acting as strain markers.

If considering a Maxwell-type rheology for the intruded crust, rapidly applied stresses, as occurring during dyke emplacement, would be accommodated by elastic processes and more slowly applied stresses would cause ductile flow. This fits well with our observations that the tip-propagation likely was driven by mode I, tensile fracturing, as this can produce embrittlement of the ductile crust and even seismogenic deformation[24], whereas somewhat slower thickening and magma ingress/pumping can cause ductile flow if the host rock is sufficiently weak.

Ductile deformation accommodating dyke emplacement implies an increasing likelihood of aseismic conditions specifically during the dyke inflation event. Interestingly, earthquakes accompanying dyke propagation at, e.g. Icelandic volcanoes, are commonly restricted to a depth range of 3-8 km[3], suggesting that laterally propagating dykes are spatially restricted to the same depths[5]. Deeper seismicity has also been documented in subvolcanic systems in Iceland[55] and in other parts of the world, such as La Réunion[56] and Mayotte[57], but is commonly associated with more complex sub-volcanic processes. Our study suggests that ductile deformation accommodates dyke inflation processes in the ductile crust, implying that dyke propagation in this part of the crust may be seismically depressed at depth.

The field data show strain gradients decreasing away from the dykes, indicating that stresses imposed by dyke inflation are accommodated by fast ductile flow along with folding and shear deformation of the host rock, consistent with a transiently weak crust. These observations have implications for interpreting geodetic data monitored at volcanoes. The most common models used to invert geodetic data associated with the emplacement of dykes assume the opening of a rectangular plane within a perfectly elastic host[58]. The surface expression of dyke emplacement is dependent on the rheology of the host rock[59], and thus our observations that dykes emplaced in a hot and weak crust, similar to Iceland and the East African Rift[40,60-62], can be associated with up to 40% ductile deformation in the host rock, should be accounted for when inverting the geodetic data.

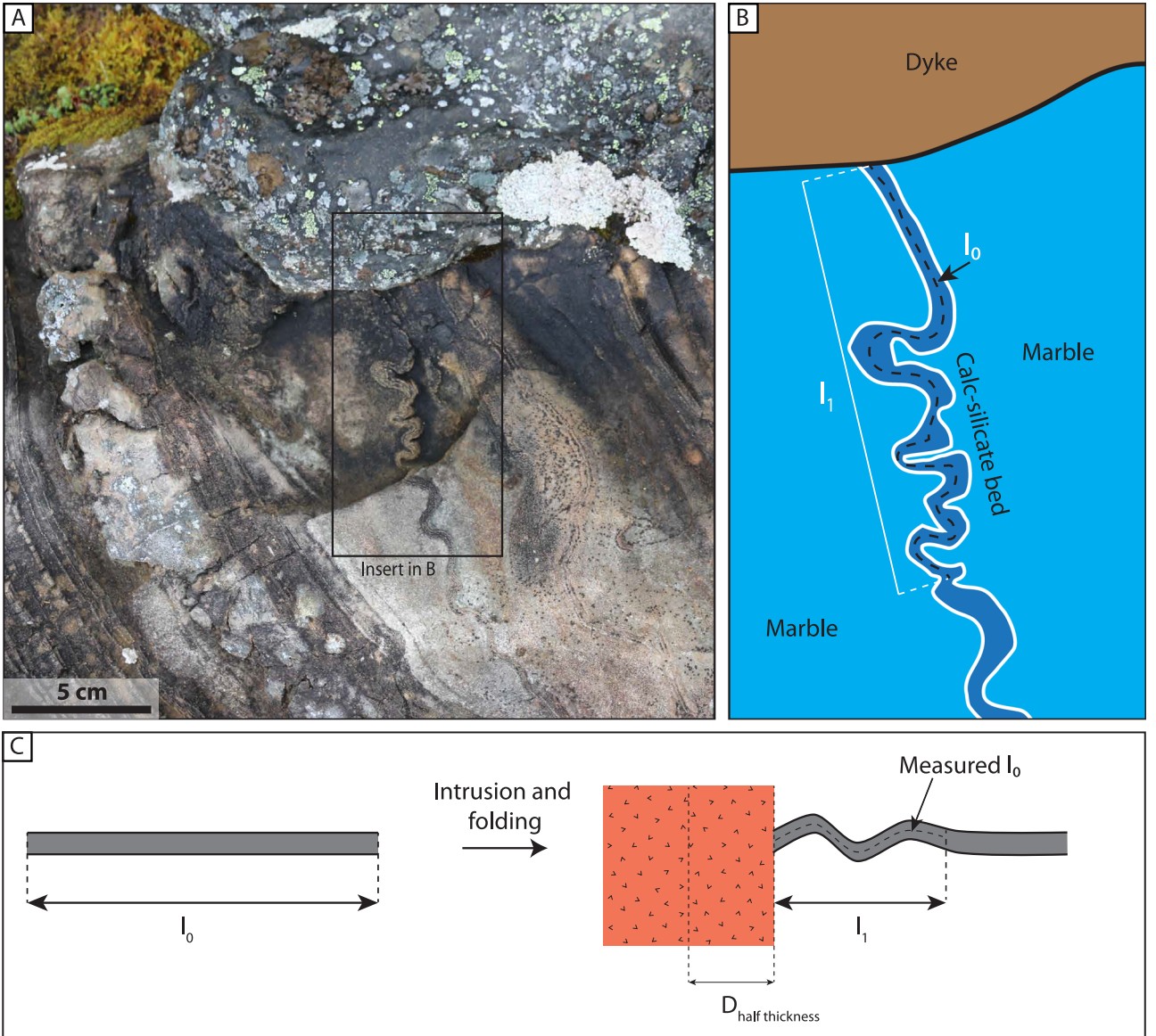

**Fig. 9 | Conceptual sketch of measurement of ductile strain compared to dyke thickness. A** Field photograph of a folded calc-silicate bed at the contact with a mafic dyke. **B** Measuring the centerline of the calc-silicate bed gives a minimum estimate of the length of the bed prior to dyke intrusion, $l_0$. The Length from the dyke contact to the point where measurement stops is the deformed length $l_1$. **C** We assume that the folding is a direct consequence of the inflation of the dyke. By measuring the half thickness of the dyke and comparing it to the one-sided deformation, we derive an estimate of how much strain was accommodated by ductile folding compared with the dyke thickness.

## Methods

The time-dependent cooling of the center of a mafic dyke of thickness x can be calculated by the following equation[27,63]:

$$T_{solidus} = T_{host} + \left(\frac{T_{magma} - T_{host}}{2}\right) 2\,\mathrm{erf}\left(\frac{\frac{x}{2}}{\sqrt{4t\frac{K}{\rho C^*}}}\right) \quad (3)$$

Where $T_{solidus}$ is the solidus temperature at the center of the dyke, $T_{host}$ is the ambient host rock temperature, $T_{magma}$ is the magma temperature during emplacement. Time t is in seconds, K is the thermal conductivity, $\rho$ is density and $C^*$ is the specific heat capacity of the magma that accounts for latent heat in the following equation:

$$C^* = C + \frac{L}{T_{liquidus} - T_{solidus}} \quad (4)$$

Here, C is the specific heat capacity of the magma, L is the latent heat of fusion, $T_{liquidus}$ is the liquidus temperature of the magma.

Strain is calculated from field photographs taken parallel to the fold axes (Fig. 9). Where permitted, average values of strain are calculated where multiple folds or multiple folded beds occur in one photograph. Dyke thicknesses are calculated from virtual 3D outcrop models generated from georeferenced photographs acquired using a drone. The virtual outcrop model was processed from 269 photographs and covers an area of 5281 m², with a pixel resolution of 2.16 mm/pixel.

Permission to conduct fieldwork in Sarek National Park was granted by the Norrbotten County Administrative Board. Authorization for helicopter and drone flights was provided by the Swedish Transport Agency.

## Data availability

The field photographs are available from figshare.com: https://doi.org/10.6084/m9.figshare.30665963.

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

## Acknowledgements

The fieldwork and HJK's position are funded by the Research Council of Norway through the Beyond Elasticity project (grant 334654). Sascha Zertani and Fabian Barras are thanked for commenting on an early version of the manuscript. Swedish authorities are acknowledged for giving permission to conduct fieldwork within Sarek National Park.

## Author contributions

H.J.K.: Fieldwork planning and execution. Data analyses and calculations. Conceptualization. Writing original draft and editing O.G.: Fieldwork planning and execution, conceptualization, review and editing, funding acquisition T.S.: Fieldwork execution, reviewing and editing.

## Competing interests

The authors declare no competing interests.
