## [Transparent Peer Review file · Nature Communications]

Rapid viscous flow of crustal rocks controls dyke emplacement in the ductile crust

Corresponding Author: Dr Hans Jørgen Kjøl

Version 1:

Reviewer comments:

Reviewer #1

(Remarks to the Author)

Dear Editor, dear authors,

I have read with interest the manuscript "Fast ductile flow controls dyke emplacement in the deep crust" submitted by Kjøl, Scheiber & Galland. The paper presents a detailed field-based study of mafic dykes intruding exhumed middle crustal host rocks in Scandinavia. The outcrop described offers a rare and exceptional natural laboratory for investigating dyke emplacement, geometry, and associated deformation and stress conditions. It is a highly valuable field site.

To my knowledge, this is the first field documentation of ductile deformation developed along dyke walls, supported by quantitative strain measurements. The authors use these observations to estimate timescales of ductile deformation during dyke emplacement, providing an important advance in linking field evidence with deformation processes.

The study offers new insights into the interplay between dyke intrusion and host-rock rheology, particularly the conditions under which weak, carbonate-dominated layers can accommodate significant ductile deformation. These observations represent a valuable complement to the extensive body of dyke-modelling studies, which typically assume an idealized elastic host.

The manuscript is well written and generally well structured, with the integration of field data and quantitative analysis being a particular strength. At the same time, it could be further improved by clarifying certain methodological and interpretive points, enhancing the integration between results, discussion, and figures, and ensuring consistency in terminology. In addition, the manuscript would benefit from improvements in clarity and accessibility, especially for readers who are not specialists in Scandinavian geology or in the microscopic analytical techniques employed. Simplifying some of the geological context or introducing more the different structures and the rationale for the microscopic analysis would help broaden the paper's reach and make the arguments more accessible to a wider volcanological audience.

Addressing these issues will sharpen the manuscript's key messages and improve its accessibility. Please find all my comments, suggestions, and questions in the attached document.

In summary, this is a novel, timely, and important contribution that will interest a wide readership. I recommend moderate revisions, as I am confident that with clarifications and some light restructuring, the paper will make a significant impact. It will be of interest to the wider volcanological community concerned with dyke propagation, magma-rock interaction, and the mechanical behavior of the crust.

Sincerely,
Séverine Furst

[Editorial Note: Please also see attachment at end of file]

Reviewer #2

(Remarks to the Author)

Dear authors,

thank you very much for letting me read this interesting article regarding the field of deeply emplaced dykes titled: Fast ductile flow controls dyke emplacement in the deep crust.

It was a pleasure to read the field approach to an otherwise often solely geophysical approached topic. While reading the text I did come across a few minor inconsistencies and aspects which I think should be made more clear. Please see the attached PDF for the respective comments in detail.

Best regards,

Tobias Schmiedel

Version 2:

Reviewer comments:

Reviewer #1

(Remarks to the Author)

Dear Editor, dear authors,

I have carefully evaluated the revised manuscript and confirm that all comments from the previous review round have been thoroughly addressed. The authors have significantly improved the figures, which now provide clearer and more accessible interpretations.

Overall, the manuscript is compelling and offers a valuable new perspective on ductile deformation that, in my view, merits consideration in certain cases of dyke propagation. This work has the potential to open new avenues for future modelling efforts and associated surface deformation studies.

I have noted a few typos and minor suggestions, which are open and do not require my approval for publication. Based on the revisions, I am satisfied with the improvements made.

Sincerely,

Séverine Furst

Line-by-line minor comments

- Line 76: "making **them** a window into"
- Figure 2: Since the captions for Fig. 2B and Fig. 2C are the same, would it be better to keep only one caption and specify this accordingly? Another suggestion: "Geological cross section after⁴⁰ along the Sarektjåhkkå and Apha Ridge (**X-X' cross section on B**)"
- Caption Fig.3: "lower hemisphere **stereoplots**". Should the yellow rectangles referring to Fig. 3A and 3B actually refer to Fig. 4A and 4B?
- Lines 147: "twins crossing the sub-grain boundaries are also bent **and/or** kinked".
- Line 334-336: Would these sentences perhaps be more appropriate in the Acknowledgements section?

General comments by reviewer 1:

Terminology consistency: In the title and abstract, the authors refer to the *deep crust*, while in the main text they use *middle crust*. I suggest being consistent throughout. It would also help readers if the authors briefly provide approximate depth ranges for shallow, middle, and deep crust (with shallow crust being the brittle domain, if I understand correctly). In addition, the authors have identified that ductile flow was observed in *weak* host rock — it may be worth highlighting this more explicitly in the abstract and introduction.

In the revised version of the manuscript, we are now consistently using the term “ductile crust” instead of “middle crust”. We have also added a word in the abstract and a sentence in the introduction highlighting that the host rock was in fact weak during the time of emplacement, as suggested by the reviewer. Depth ranges are shown in Fig. 1 together with the terms brittle crust, brittle-ductile transition and ductile crust.

Geological setting: The geological background is interesting and detailed, but may be challenging for readers unfamiliar with Scandinavian geology. If the authors wish to appeal to the broader dyke modelling community, they could consider simplifying or clarifying some parts. For example, a brief explanation of “Baltica” would be valuable.

Following the reviewer’s comment, we have added some words and sentences explaining the niche words used such as “Baltica” and “Scandian phase” (see line numbers 70-73). In addition, we have simplified and added supporting sentences and phrases in order to make the text more digestible for a broader audience.

Microscopic analysis: While potentially interesting, the microscopic analysis is currently underdeveloped. As it stands, the short section and figure do not integrate well with the rest of the manuscript and do not appear to support later arguments. If the authors wish to retain it, I recommend they provide a clearer rationale for including it, and explain how it contributes to the overall findings.

We have added a sentence (Line 143) before we introduced the results from the microstructures to highlight why these observations are important. We have also updated the text in the paragraph to emphasize the comparison between the deformed and undeformed host rock. Further we have added a specific point in line 157 and 158 to show the relevance of the microstructural interpretation on our conclusion that the folding is directly related to dyke emplacement.

Ductile vs. non-ductile contacts: Some contacts between dyke and marble dominated host rock display ductile deformation, while others do not (e.g., Fig. 4C). Could the authors comment on why this is the case? Why might it occur on one side of a dyke and not the other? I noticed this asymmetry is addressed later in the discussion; perhaps a brief mention in the results, with a cross-reference to the later section, would help guide the reader.

We acknowledge this comment and have added a sentence and a figure reference on line 127-128: “Locally, folding is asymmetric such that the host rock is folded on one side of the dyke and not the other (Fig. 5B).” We have also added a sentence in the figure caption of figure 5: “The asymmetry of the system is likely related to the highly

heterogeneous nature of the system”. Furthermore, we added a sentence and a reference directly addressing this issue in the discussion on line 235-237: “The occurrence of one-sided folding of the host rock is likely controlled by local stress disturbances caused by the inherently heterogenous nature of dyke emplacement processes (Souche et al., 2019)”

Strain measurements: The authors state they measured 11 longitudinal strains (line 165) and provide one example in Fig. 10, with the full dataset included in the supplementary material. It would be clearer if the supplementary material were explicitly referenced in the main text. The accompanying photographs could also be placed in the supplementary alongside the table.

Regarding terminology: when referring to *symmetric folds*, do the authors mean symmetry relative to the dyke itself, or symmetry of the folds’ geometry? Clarification here would help.

We acknowledge the need for a better data presentation and have decided to include an expanded table from the supplementary material (sup. 1) in the main manuscript as table 1. We make sure to cross reference the table where we present the data in the text. We have also updated figure 5 with an extra field photograph such that most of the folds are actually shown in the figure. These folds are cross-referenced to table 1.

Regarding terminology, we are referring to folds that are symmetric with regards to the geometry of the fold. This is now specified in the text on line 171. When we are referring to the symmetric or asymmetric folding around a dyke, we have added some words specifying this in the text. E.g. figure caption of figure 5, line 127-128 and line 231-234.

Half-width and shortening measurements: These seem to have been taken at different positions along the dyke — narrower near the tip and wider farther away. Do the authors observe a relationship between dyke position and the degree of ductile deformation?

Mapping ductile vs. non-ductile contacts along dyke walls could clarify whether deformation initiates at the tip or occurs elsewhere, and whether it becomes more pronounced closer to or farther from the tip. Although this point is addressed later (lines 222–226), an earlier cross-reference in the results would improve clarity.

To clarify, we observe folding along numerous dyke contacts. We have updated the first sentence in the result chapter “Deformation features near dyke walls” to highlight this fact (line 117).

As stated in the discussion, we do see that thinner dykes have a greater percentage of deformation compared to their thickness than thicker dykes. However, our dataset is too sparse to draw a robust conclusion and that is why this is mentioned briefly in the discussion and not elaborated on extensively. We also mention this in the end paragraph related to thickness vs. shortening in line 183-184: “Thin dykes, such as the dyke in figure 5B, has the highest amount of strain compared to the total thickness (Tab 1)”.

Note as well that no dyke tip is exposed at the studied outcrop, thus it is very challenging to estimate the distance to the tips of the studied dykes. In the

manuscript, we enhance the discussion whether the observed ductile deformation is related to tip processes or processes along the dyke walls.

Cross-section geometry: Could the authors clarify whether the measured dyke widths are true widths, unaffected by the orientation of the outcrop? If the crosssection is oblique (cut along strike rather than breadth), the apparent width could be exaggerated.

All the measurement of dyke thickness were done orthogonal to the dyke contact, some in the field and some on the 3D virtual outcrop model, so the stated thickness is the true thickness of the dyke to the best of our abilities. This is now mentioned in the manuscript at line 111.

Strain rate estimates: The dykes in the study area range from 1–10 m in thickness, but folds are only documented in dykes up to 5 m thick. Were no folds observed in the thicker dykes, or were they inaccessible? If thicker dykes do not show folding, this could imply that ductile deformation is limited to smaller intrusions. In that case, why do the strain rate calculations use an opening range up to 10 m? The authors might consider justifying this choice, i.e. clarifying why the 5–10 m range is included as an upper width boundary, resulting in a lower strain rate boundary.

This is an excellent question and, as also stated in the rebuttal letter above, we chose the values based on previously published work (Kjøll et al., 2019 - <https://doi.org/10.1016/j.epsl.2019.04.016>). During the preparation of the revised version of this manuscript, however, we realized that we in fact could calculate the strain rate for each individual fold in the data set, as we know the dyke thickness for all of the dykes at the place where folding occurs. We consider this approach to be more robust and data-driven and have therefore chosen to implement the method.

We have added a sentence about the dyke thickness distribution in all of Sarek in the geological setting (line: 92 – 95) and a sentence about the dyke thicknesses in the study area (line 110-111).

Timing of folding: The paragraph in the discussion that addresses the timing of folding relative to dyke emplacement (lines 212-221) is very insightful and central to the interpretation. I suggest moving it into the results section, as it strengthens the flow leading into the timing and strain-rate estimates.

We agree that this would strengthen the flow, therefore we have included some sentences in the results chapter prior to the strain and strain rate sub-chapters to strengthen the flow as suggested by the reviewer. Line 163-166: “The exact timing of the deformation is challenging to derive, but the observations of folding along the dyke wall close to dyke tips (Fig. 5B) as well as the correlation of dyke step geometries and folded host rock (Fig. 5E), suggest that folding occurred early during dyke inflation while a significant part of the dyke interior still was molten.”

Depth of dyke emplacement: The manuscript states that earthquakes during dyke propagation in Iceland are typically restricted to 3–8 km depth. Could the authors specify here why they are taking Iceland as an example? Indeed, deeper seismicity has been

recorded in other volcanic unrest (e.g., Piton de la Fournaise, Battaglia et al., 2003; Mayotte, Mercury et al., 2023). It may also be worth emphasizing that hostrock rheology — weak vs. strong layers (e.g., marble vs. calc-silicate) — strongly influences whether ductile deformation occurs.

In the cases brought forward by the reviewer, the seismicity is indeed related to volcanic unrest, but much more complicated systems related to central shield volcanoes, where the deeper seismic activity likely is related to processes such as decompression of deep magma reservoirs. The case reported by Augustdottir et al. (2015), on the other hand, is more clearly related to the 48 km propagation of a single dyke leading to the 2014 Bardarbunga-Holuhraun eruption. To account for this, we have added the words “laterally propagating” in line 298. As well as rewriting the final part of the paragraph from line 399-33:

“Deeper seismicity has also been documented in subvolcanic systems in Iceland (Greenfield et al., 2022) and in other parts of the world such as La Reunion (Battaglia et al., 2005) and Mayotte (Mercury et al., 2022) but is commonly associated with more complex sub-volcanic processes. Our study suggests that ductile deformation accommodates dyke inflation processes in the ductile crust, implying that dyke propagation in this part of the crust may be seismically depressed at depth”

Surface deformation implications: If 14–40% of deformation is accommodated by ductile deformation in weak layers, what are the implications for surface deformation signals? Would this cause over- or under-estimates of dyke-induced ground deformation? Could the authors provide an order-of-magnitude estimate, or at least a qualitative discussion? Since these processes occur at middle crustal depths, would they be detectable in geodetic observations?

This is a very important point and we have added a sentence to further underpin the importance of this issue. Estimating the effects of dyke-induced ductile deformation on surface deformation would require the development of new mathematical codes for geodetic models, which is beyond the scope of our field-based study. The last paragraph in the discussion chapter “Implications for dyke emplacement and volcanic rifts” deals with qualitative implications of our findings for geodesy and we have added the following sentence on line 309-311: “The surface expression of dyke emplacement is dependent on the rheology of the host rock (Bertelsen et al., 2021)”

References in support of hot/weak middle crust: Citations are needed for the statement that the middle crust beneath Iceland and the East African Rift is hot and weak, comparable to the study area.

We have added several references to support this statement in line 311.

Cooling-time equation: The equation for the characteristic cooling time of a mafic dyke should be referenced. Additionally, why is dolostone used as the “reference” rock for this calculation? A brief justification or citation would strengthen the methods.

We have added two references for the classical cooling-time equation. Further we have removed the text regarding the dolostones as this was left over from a previous version, that is no longer relevant to the study.

Estimation of ductile strain: The authors compare dyke half-width with one-sided ductile deformation to estimate accommodated strain. However, if deformation is asymmetric (occurring on one wall but not the other), is this approach still valid? Some clarification here would be useful.

Because of the partly-scrub covered nature of the outcrop this is challenging to constrain accurately, but host rock deformation has been observed generally on both sides of the dykes. In some few cases, however deformation seems to be restricted to one side only. But this is rather the exception. Note however that asymmetric ductile deformation would affect the results by a factor ~2, thus in the same order of magnitude, which would not modify our conclusions.

Figure-specific comments

- Fig. 1: In the right panel, should the label read “Thermally weakened crust”?

Correct. This issue has been fixed.

- Fig. 2: The figure is very useful but could be made easier to follow.

- o Clarify whether panel B corresponds to the square in panel A.

Done – it is the location of the national park.

- o Are the grey lenses glaciers (as in panel C) or ophiolite/volcanic arc assemblages (as in panel A)?

Done – they are glaciers

- o Transitioning from panel B to C is currently unclear; additional description or topographic contours could help.

We have added a hillshaded basemap to highlight the topography.

- o The authors should specify the location of the cross section shown in C, in panel B.

Indeed. It is now marked on the map with an x to x'

- o What does the actual NW-SE line represent?

This is the location of the cross-section. It is now marked.

- o It would be valuable to locate the cliff and viewpoint within panel B.

It is now indicated with an arrow.

- o Would the authors consider having a unique geological scale for panel B and C? At the moment there are repetitions and color duplicates with different captions.

Most of the rock units in B and C are the same. We have adapted the colors, changed the extent of the cross-section, and changed the names in the legend to make this clearer.

- **Fig. 3:** Clarify whether this photo represents the entire outcrop or a subset. Specify how many dykes are included on the photo. If multiple localities were studied, they should be presented in Fig 3 and shown on Fig. 2E. In the caption: “Yellow rectangles highlight locations of figures presented below Fig.4 and 5.”

Figure 2E is an overview to show the local exceptional outcrop quality. We have now made that clear in the figure caption. We have also clarified that Figure 3 displays the central part of the studied outcrop. The rest is not included simply for practical reasons since it is mostly scree covered and the extent would be too large to show clear dyke-host rock relationships.

- Fig. 4: Could panels B and C be located within Fig. 3A, as was done for Fig. 5? Is the close-up view in panel A taken from somewhere on Fig 4A? In panel C, why does the marble contact show no ductile deformation?

We have updated figure 3 in accordance with the reviewer’s comment. For the last question, this is from a part of the outcrop where there is not recorded two-sided deformation, but the deformation is localized on the other side of the intrusion.

- Figs. 4 & 5: Consider adding dominant lithology labels directly on the photos.

We have added some more annotations explaining the lithologies.

- Fig. 5: Define ϵ (strain) clearly in the caption and link it to the ductile strain-rate section.

Done

- Fig. 8: “c.f.” before the reference seems unnecessary.

Removed

- Fig. 9: Consider making the blue gradient in panels B and C more pronounced.

Done

- Fig. 10: To remain consistent, use “marble” rather than “carbonate.” In the caption, refine the description of L1 using the description from the main text: “*L1 is the distance from the dolerite–host rock contact to where the deformation fades.*” This should also be corrected in panel B, where currently L1 stops before the deformation fades.

Changed

Line-by-line comments:

Line 15 and 59–60: Please specify that the 27% value represents an *average* of dyke thickness. This value is not present in the “Ductile strain rate estimate” section, where only a range is provided. Consider adding the average value alongside the range.

We have now added the average value in the “ductile strain rate estimate” subchapter, line 179.

Lines 57–64: During my first reading of this paragraph, I felt that the information was presented somewhat abruptly, as if several points were simply listed without sufficient connection. I suggest adding a linking sentence that explicitly highlights the key observation

— namely, that folds are observed along the dyke walls in weaker host rocks. This would help guide the reader and give the paragraph a clearer focus.

We have now added the sentence: “The dyke complex was emplaced in thermally weakened marbles and arkoses. Folds in the weak host rocks, with axial planes sub-parallel to the dyke contacts, are documented along the dyke walls.” To clarify this issue.

Line 59: “27% of the dyke *inflation*”. I would recommend the authors to keep *inflation* for describing the processes as there are doing later in the text, and use *thickness* here as they describe it in the abstract and results sections.

Agree, changed

Lines 91–92: References should be formatted as superscript numbers.

Changed

Line 128: From Fig. 5, it is not possible to verify the statement that “thicker dykes show more intense folding.” Could clarification or supporting evidence be provided?

We see the confusion and have tried to clarify by changing the sentence to: “Where the dyke steps into the host rock,…”

Lines 136–138: Two questions arise:

1. Maybe this is a naive question, but under what conditions can shortening occur non-perpendicular to the dyke contact?
If the folding was associated with e.g. later tectonic shortening of the area one could envisage that the shortening could be non-perpendicular to the dyke wall and accommodate significantly more shear.
2. I imagine that the present outcrop exposure may not reflect emplacement conditions (e.g., strike/dip may differ). Could the observation from the author give an indication about the orientation of the dyke during emplacement and in turn inform about the past stress conditions?
This is an interesting point, which has been described in a previous publication from the area (Kjøll et al., 2019). The area has suffered an unknown rotation after the dyke emplacement and possibly also pre-emplacement (c.f. syn tectonic rotation of hanging-wall rocks in rifted margins). Another complicating factor is that the dykes are not straight, but rather inclined and show a zig-zag pattern, making such a paleo stress study challenging. The paper mentioned above suggest three possible paleo-reconstructions, however, none of them are fully conclusive.

Line 168: The strain values (2.7–65.2%) could be more effectively presented by adding them as a color scale on Fig. 7A. Ensure Figs. 7A and 7B are both referenced in the text; Fig. 7B is not described nor mentioned in the text.

This is a good point and we have now color coded the datapoints in accordance with the suggestion of (both) reviewers.

Line 176: Should refer to *Fig. 8*, not Fig. 7.

Changed

Lines 184–187: Why are ranges/mean/median given for only one end-member?

This has now been updated and we provide ranges and averages for all. Since we now use the actual data to calculate the strain rates, we now give the maximum and minimum value.

Line 196: Replace comma with a period: "...at the dyke tip. Therefore, ..."

Done

Line 211: Correct "smalls trains" → "small strains."

Done

Line 244: Suggested rephrasing: "*Where the host rock is strong, i.e., dominated by calc-silicate beds, it resists magma intrusion.*"

Changed

Line 295: The reference to Ji et al. could be complemented with "e.g." to indicate it is one among several modelling approaches.

It is actually, but endnote did not allow the "e.g." to be in superscript. I suppose this will be formatted correctly in a possible typeset version.

References: Ensure consistency — either always list all authors or use "et al." consistently.

The references are now updated in accordance with Nature communications guidelines.

Answers to reviewer number 2

The smaller comments addressing grammatical errors and typos have been fixed without further mentioning in this document.

Line 15: I highlighted this number in a different colour similar to a few others in the text.

I am not completely sure, where this number comes from, yes it is mentioned in the text once more (Line 59). However, when going to the part with the "Ductile strain rates estimate

section", the shortening average calculated to 23% (Line 164). Is that the same number, if not please make it more clear where the difference lies and where the 27% are actually calculated.

This was a mistake, where a previously calculated number had not been updated. It is now corrected to 25%, throughout the manuscript.

Line 107: Is this the same direction as the described NE-SW in the figure caption of figure 3? If yes, please make that clear

It is and the figure caption is now updated with the correct strike direction.

Line 152: I am not sure I understand what the authors meant to say here. Do you mean compressing/shortening of the host rock due to inflation? Without any shear caused by the emplacing intrusion?

In any case it seems to me that the expression in the figure caption stand in contrast to what is written in the text (Line 215-217). Could the authors please clarify this mismatch?

We observe disharmonic folding along the dyke wall, this folding is interpreted to have been formed during pure shear shortening of the host rock adjacent to the dyke. We have changed the figure caption to now say: "disharmonic folding formed during pure shear deformation of the host rock". We do not find that this statement is in contrast or conflict to what is written in line 215-217, as this sentence also state that the folding is accommodating pure shear deformation.

Line 153: Wonderful microphotograph and nice quality of the microscopic images, thank you for that.

Are the thin section oriented? It would be great to relate the direction of deformation on the subgrain scale to the larger direction of deformation of the host rock.

Unfortunately, this specific thin-section is not oriented as it was a complicated piece to get loose.

Line 170: I am not sure I understand these plots correctly. The dyke half width is the combining feature, correct? Therefore, I would expect the x values in both plots to be the same. That is not the case as i tried to indicate with the red lines.

Could the authors please address this mismatch or clarify what I am potentially overlooking? Thank you in advance

Could colour coding the samples potentially be a solution, given the relatively small amount of dots?

The mismatch between the x-axis values in an unfortunate editing artefact. It has now been corrected. In the revised version we have added a color coding to the datapoints, in accordance with the comments from both reviewers.

Line 215-217: I mentioned this in one of the early comments: Is this statement not opposing to what is said in the captions of figure 5: "opposite facing directions and pure flattening"?

In this context maybe also Figure 9C should be double checked for clarity on this matter

Please clarify.

To avoid confusion, we now specify that the observed folds are often symmetrical or disharmonic with axial planes parallel to the dykes and that this suggests pure shear deformation.

Line 223-224: Could a colour coding of the samples make this relation more visible in figure 7?

Absolutely, and this is now implemented in the scatter plot.

Line 224-226: As it stands this is a rather vague formulation.. To strengthen this statement it would be great if the authors could provide a potential underlying physical or chemical process which would lead to such an observed behaviour of initial high deformation compared to later stage timing. To give the reader a better grasp on why to trust the proposed suggestion?

We think that after adding the color coding of figure 7, it becomes much clearer that the thin dykes have accommodated the most strain compared to the thickness of the adjacent dyke. The actual processes behind this observation, however, are more complicated, and we think that it would be too speculative to elaborate on this issue in this contribution.

Line 266: I think, this might simply be a matter of rephrasing. At the moment, it reads to me that this sentence is in contrast to the sentence before, with respect to the strength of the calc-silicate layers.

Please clarify.

Yes, we see the confusion and have tried to make the sentence clearer by adding some words: "Disharmonic folding of thin calc-silicate beds indicates the overall weak nature of the host rock".

I suggest rephrasing this for two reasons. The references never talk about explicitly about "blade-shape" and the last part reads to me as if it misses a "word" like: "...propagate from deeper/greater depth than previously assumed."

Please change that for better clarity

We decided to remove the last part of the sentence to avoid ambiguities.

Line 314: Why parameters for dolostone here and in the literature table the parameters and values for basalt? Where is the value used for the dolostone mentioned?

Please clarify this.

This was left over from an earlier version of the manuscript and is now updated.

Line 318-321: What are the resolutions and potential errors of those 3D outcrop models? Given that the authors mention a shortening representing down to 2.7% of the dyke thickness (1-5 m of measured dyke thickness). How reliable is that number?

We have updated the manuscript with some statistics of the model by adding the following sentence: “The virtual outcrop model was processed from 269 photographs and covers an area of 5281m², with a pixel resolution of 2.16 mm/pixel.”

Line 325: It would be nice to format the references, some are easy to follow through the links, some only provide the doi as text, some are missing the doi/links.

Please be consistent with that.

Yes, the formatting of the references are updated in accordance with Nature Communications standard.

General comments

- **Terminology consistency:** In the title and abstract, the authors refer to the *deep crust*, while in the main text they use *middle crust*. I suggest being consistent throughout. It would also help readers if the authors briefly provide approximate depth ranges for shallow, middle, and deep crust (with shallow crust being the brittle domain, if I understand correctly). In addition, the authors have identified that ductile flow was observed in *weak* host rock — it may be worth highlighting this more explicitly in the abstract and introduction.
- **Geological setting:** The geological background is interesting and detailed, but may be challenging for readers unfamiliar with Scandinavian geology. If the authors wish to appeal to the broader dyke modelling community, they could consider simplifying or clarifying some parts. For example, a brief explanation of “Baltica” would be valuable.
- **Microscopic analysis:** While potentially interesting, the microscopic analysis is currently underdeveloped. As it stands, the short section and figure do not integrate well with the rest of the manuscript and do not appear to support later arguments. If the authors wish to retain it, I recommend they provide a clearer rationale for including it, and explain how it contributes to the overall findings.
- **Ductile vs. non-ductile contacts:** Some contacts between dyke and marble-dominated host rock display ductile deformation, while others do not (e.g., Fig. 4C). Could the authors comment on why this is the case? Why might it occur on one side of a dyke and not the other? I noticed this asymmetry is addressed later in the discussion; perhaps a brief mention in the results, with a cross-reference to the later section, would help guide the reader.
- **Strain measurements:** The authors state they measured 11 longitudinal strains (line 165) and provide one example in Fig. 10, with the full dataset included in the supplementary material. It would be clearer if the supplementary material were explicitly referenced in the main text. The accompanying photographs could also be placed in the supplementary alongside the table. Regarding terminology: when referring to *symmetric folds*, do the authors mean symmetry relative to the dyke itself, or symmetry of the folds’ geometry? Clarification here would help.
- **Half-width and shortening measurements:** These seem to have been taken at different positions along the dyke — narrower near the tip and wider farther away. Do the authors observe a relationship between dyke position and the degree of ductile deformation? Mapping ductile vs. non-ductile contacts along dyke walls could clarify whether deformation initiates at the tip or occurs elsewhere, and whether it becomes more pronounced closer to or farther from the tip. Although this point is addressed later (lines 222–226), an earlier cross-reference in the results would improve clarity.
- **Cross-section geometry:** Could the authors clarify whether the measured dyke widths are true widths, unaffected by the orientation of the outcrop? If the cross-section is oblique (cut along strike rather than breadth), the apparent width could be exaggerated.
- **Strain rate estimates:** The dykes in the study area range from 1–10 m in thickness, but folds are only documented in dykes up to 5 m thick. Were no folds observed in the thicker dykes, or were they inaccessible? If thicker dykes do not show folding, this

could imply that ductile deformation is limited to smaller intrusions. In that case, why do the strain rate calculations use an opening range up to 10 m? The authors might consider justifying this choice, i.e. clarifying why the 5–10 m range is included as an upper width boundary, resulting in a lower strain rate boundary.

- **Timing of folding:** The paragraph in the discussion that addresses the timing of folding relative to dyke emplacement (lines 212–221) is very insightful and central to the interpretation. I suggest moving it into the results section, as it strengthens the flow leading into the timing and strain-rate estimates.
- **Depth of dyke emplacement:** The manuscript states that earthquakes during dyke propagation in Iceland are typically restricted to 3–8 km depth. Could the authors specify here why they are taking Iceland as an example? Indeed, deeper seismicity has been recorded in other volcanic unrest (e.g., Piton de la Fournaise, Battaglia et al., 2003; Mayotte, Mercury et al., 2023). It may also be worth emphasizing that host-rock rheology — weak vs. strong layers (e.g., marble vs. calc-silicate) — strongly influences whether ductile deformation occurs.
- **Surface deformation implications:** If 14–40% of deformation is accommodated by ductile deformation in weak layers, what are the implications for surface deformation signals? Would this cause over- or under-estimates of dyke-induced ground deformation? Could the authors provide an order-of-magnitude estimate, or at least a qualitative discussion? Since these processes occur at middle crustal depths, would they be detectable in geodetic observations?
- **References in support of hot/weak middle crust:** Citations are needed for the statement that the middle crust beneath Iceland and the East African Rift is hot and weak, comparable to the study area.
- **Cooling-time equation:** The equation for the characteristic cooling time of a mafic dyke should be referenced. Additionally, why is dolostone used as the “reference” rock for this calculation? A brief justification or citation would strengthen the methods.
- **Estimation of ductile strain:** The authors compare dyke half-width with one-sided ductile deformation to estimate accommodated strain. However, if deformation is asymmetric (occurring on one wall but not the other), is this approach still valid? Some clarification here would be useful.

Figure-specific comments

- **Fig. 1:** In the right panel, should the label read “Thermally weakened crust”?
- **Fig. 2:** The figure is very useful but could be made easier to follow.
 - Clarify whether panel B corresponds to the square in panel A.
 - Are the grey lenses glaciers (as in panel C) or ophiolite/volcanic arc assemblages (as in panel A)?
 - Transitioning from panel B to C is currently unclear; additional description or topographic contours could help.
 - The authors should specify the location of the cross section shown in C, in panel B.
 - What does the actual NW-SE line represent?
 - It would be valuable to locate the cliff and viewpoint within panel B.

- Would the authors consider having a unique geological scale for panel B and C? At the moment there are repetitions and color duplicates with different captions.
- **Fig. 3:** Clarify whether this photo represents the entire outcrop or a subset. Specify how many dykes are included on the photo. If multiple localities were studied, they should be presented in Fig 3 and shown on Fig. 2E.
In the caption: “Yellow rectangles highlight locations of ~~figures presented below Fig.4 and 5.~~”
- **Fig. 4:** Could panels B and C be located within Fig. 3A, as was done for Fig. 5? Is the close-up view in panel A taken from somewhere on Fig 4A? In panel C, why does the marble contact show no ductile deformation?
- **Figs. 4 & 5:** Consider adding dominant lithology labels directly on the photos.
- **Fig. 5:** Define ϵ (strain) clearly in the caption and link it to the ductile strain-rate section.
- **Fig. 8:** “c.f.” before the reference seems unnecessary.
- **Fig. 9:** Consider making the blue gradient in panels B and C more pronounced.
- **Fig. 10:** To remain consistent, use “marble” rather than “carbonate.” In the caption, refine the description of L1 using the description from the main text: “L1 is the distance from the dolerite–host rock contact to where the deformation fades.” This should also be corrected in panel B, where currently L1 stops before the deformation fades.

Line-by-line minor comments

- **Line 15 and 59–60:** Please specify that the 27% value represents an *average* of dyke thickness. This value is not present in the “Ductile strain rate estimate” section, where only a range is provided. Consider adding the average value alongside the range.
- **Lines 57–64:** During my first reading of this paragraph, I felt that the information was presented somewhat abruptly, as if several points were simply listed without sufficient connection. I suggest adding a linking sentence that explicitly highlights the key observation — namely, that folds are observed along the dyke walls in weaker host rocks. This would help guide the reader and give the paragraph a clearer focus.
- **Line 59:** “27% of the dyke *inflation*”. I would recommend the authors to keep *inflation* for describing the processes as there are doing later in the text, and use *thickness* here as they describe it in the abstract and results sections.
- **Lines 91–92:** References should be formatted as superscript numbers.
- **Line 128:** From Fig. 5, it is not possible to verify the statement that “thicker dykes show more intense folding.” Could clarification or supporting evidence be provided?
- **Lines 136–138:** Two questions arise:
 1. Maybe this is a naive question, but under what conditions can shortening occur non-perpendicular to the dyke contact?
 2. I imagine that the present outcrop exposure may not reflect emplacement conditions (e.g., strike/dip may differ). Could the observation from the author give an indication about the orientation of the dyke during emplacement and in turn inform about the past stress conditions?

- **Line 168:** The strain values (2.7–65.2%) could be more effectively presented by adding them as a color scale on Fig. 7A. Ensure Figs. 7A and 7B are both referenced in the text; Fig. 7B is not described nor mentioned in the text.
- **Line 176:** Should refer to *Fig. 8*, not Fig. 7.
- **Lines 184–187:** Why are ranges/mean/median given for only one end-member?
- **Line 196:** Replace comma with a period: "...at the dyke tip. Therefore, ..."
- **Line 211:** Correct "smalls trains" → "small strains."
- **Line 244:** Suggested rephrasing: "*Where the host rock is strong, i.e., dominated by calc-silicate beds, it resists magma intrusion.*"
- **Line 295:** The reference to Ji et al. could be complemented with "e.g." to indicate it is one among several modelling approaches.
- **References:** Ensure consistency — either always list all authors or use "et al." consistently.